Science Science

[andreapioferreri@gmail.com]

[marilena.filippucci@uniba.it]

# The new seismic catalog of the Gargano area (Southern Italy) after a decade of seismic monitoring by OTRIONS network

Andrea Pio Ferreri<sup>1,\*</sup>, Annalisa Romeo<sup>2</sup>, Rossella Giannuzzi<sup>1</sup>, Teresa Ninivaggi<sup>3</sup>, Marilena Filippucci<sup>1,4,\*</sup>, Gianpaolo Cecere<sup>3</sup>, Luigi Falco<sup>3</sup>, Maddalena Michele<sup>4</sup>, Giulio Selvaggi<sup>4</sup>, and Andrea Tallarico<sup>1,4</sup>

<sup>1</sup>Dipartimento di Scienze della Terra e Geoambientali, Università di Bari - Aldo Moro, Bari, Italy

<sup>2</sup>Istituto Nazionale di Geofisica e Vulcanologia, Osservatorio Vesuviano, Napoli, Italy

<sup>3</sup>Istituto Nazionale di Geofisica e Vulcanologia, Sezione Irpinia, Grottaminarda (AV), Italy

<sup>4</sup>Istituto Nazionale di Geofisica e Vulcanologia, Osservatorio Nazionale Terremoti, Rome, Italy

\*Correspondence: andreapioferreri@gmail.com, marilena.filippucci@uniba.it

Abstract. The Gargano Promontory (hereafter GP) has attracted the attention of seismologists in recent years for its peculiarities regarding the high rate of low-magnitude seismicity and focal depths in the lower crust. These peculiarities have been

- highlighted thanks to the new data provided by the OTRIONS seismic network (hereafter OT), installed in 2013 in the GP area, consisting of 15 short-period seismometers, thanks to a fruitful collaboration between UniBa (University of Bari Aldo Moro) and INGV (Istituto Nazionale di Geofisica e Vulcanologia). The first available seismic catalog refers to the first 7 year of the network operation (2013-2018) suffered of some technological problems of the acquisition system. Thanks to improvements in the data transmission system, these problems were overcome in 2019 and now the OT network data are available in real time.
- In order to include the most recent seismicity and to cover the temporal gaps existing in the previous catalog, we thoroughly reviewed the 24h seismic recordings, collected over the decade after the installation, by employing an automatic detect and picking software (CASP, Complete Automatic Seismic Processor). More than 6900 seismic events were initially identified. Through careful manual review, approximately 60% were confirmed as local earthquakes, and the others were recognized as quarry blasts or false/poorly-located events. Manual revision significantly improved the quality of P- and S-phase picks, which
- led to more accurate earthquakes locations by both linearized and non-linear algorithms obtaining 2 catalogs both here released. This study highlights the value of semi-automated analysis for seismic catalog compilation but manual revision is still necessary. The quality of the catalogs was assessed in detail using statistical parameters and a new formula for the location quality. The completeness magnitude of the new catalogs is as low as 0.82. The noise affecting the network was also evaluated. This study confirms the importance of the OT local network for seismic hazard analysis and provides a useful data set for
- seismotectonic and geophysical studies in a long under-monitored region.

# 1 Introduction

Monitoring small to moderate magnitude earthquakes is essential to understand the seismotectonic, seismic hazard and strongmotion characteristics in region unexplored and to this end, the installation of dense seismic networks is essential. It is observed that the increasingly dense coverage provide better data quality that, together with novel technologies in observational seismology, have significantly improved the earthquake detection capabilities worldwide (refer to Li (2021) for a review).

Of particular interest for Southern Italy are the examples of installation of permanent or temporary local seismic networks to improve the detection threshold of the Italian National Seismic Network (RSN) managed by INGV. In the followings, some example of local seismic monitoring in southern Italy are reported. The Irpina Seismic Network (ISNet) was deployed in 2005 and provided valuable information on the seismicity of the Southern Apennines region, contributing to the understanding

- of seismogenic processes in an area hit by the strong earthquake of 1980, November 23, Ms = 6.9 earthquake, in a period in which the coverage of RSN was very poor (Iannaccone et al., 2010). On the island of Ischia (Southern Italy), the results of the local seismic network contribute to research on volcanic and seismic risk assessment, providing data that can inform disaster preparedness and response strategies and has been significantly improved after the earthquake of 21 August 2017  $(M_d = 4.0)$ , that caused two fatalities, (Tramelli et al., 2024) and it was fundamental to record and constraint the few and
- small ( $M_{max} = 2.5$ ) aftershocks. In the Val d'Agri (Southern Italy) the local seismic network INSIEME, composed of seismic stations equipped with broadband posthole sensors positioned in wells with depths from 6 m to 50 m, is functional to monitor the microseismicity induced both by hydrocarbon extraction and by activities related to the water supply of the region (Stabile et al., 2020). In the Pollino region, several seismic networks have been operational, following a 4-year increase in seismicity occurred in 2010. During this period, a group of temporary seismic stations from the INGV (Govoni et al., 2023) and the
- Deutsches Geo Forschungs Zentrum (GFZ) (Passarelli et al., 2012, 2017) (Passarelli et al., 2012) were installed in order to improve the real-time earthquake identification capability of the Italian National Seismic Network. De Gori et al. (2022) describe how the Pollino seismic networks were essential to detect and analyze the seismicity and contributed to the study of the deformation dynamics highlighting the presence of an active and complex fault system.

From the examples above, without claiming to be exhaustive with respect to the wide diffusion of local networks in Italy 45 and worldwide, it can be argued that the multiplication of seismic stations involves, as a consequence of the large amount of seismic data, a strong effort in terms of human and information technology resources and therefore also raises the question of a non-trivial increase in costs and it is precisely the trade-off between scientific benefits and costs that, according to Ebel (2008), would requires a broader public debate even though it is indubitable the improvement in the seismological knowledge for hazard mitigation and public safety issues. Therefore, earthquake monitoring is the basis of observational seismology and

50 earthquake catalogues are its main product, essential in all the seismological based studies. The amount of continuous data available 24h moves the seismological community toward automated processing approach to the arrival time picking of P and S waves and to earthquake location.

In this paper our focus is on the GP area that belongs to the Apulia region (Southern Italy) which is part of the Adria plate, shown in Fig. 1a (refer to Del Gaudio et al. (2007); Pierri et al. (2024) for a review). GP represents the northernmost sector of

**Figure 1.** a) Brown area represents the Adria microplate with the studied area of the GP in the red square. b) Generalized geological map of the GP area highlighting the principal stratigraphic sequences and fault systems.

- the Apulian foreland and is predominantly composed by slightly deformed carbonatic successions (Del Gaudio et al., 2007). A generalized geological map, in which the principal stratigraphic sequences and fault systems of the GP area are represented, is shown in Fig. 1. Several studies have identified some of the major faults, shown in Fig. 1b, like the Mattinata fault (MF) (Chilovi et al., 2000), the Apricena fault (AF) (Patacca and Scandone, 2004), the Candelaro fault (CF) (Mongelli and Ricchetti, 1970), and the Sannicandro fault (SF) (Salvi et al., 1999), although their characteristics and evolution still remain under discussion.
  In addition, according to Doglioni et al. (1994), the GP is also characterized by a parallel fault system with an east-west orientation, causing uplift in this region. Six strong earthquakes throughout history, with magnitudes greater than 6, struck the GP area, the biggest of which in 1627 with M<sub>w</sub> = 6.7 (Rovida et al., 2016) while, in the instrumental era, the seismicity of GP is characterized by low magnitude seismicity with high seismic rate. A M<sub>w</sub> = 4.8 earthquake occurred in March 2025, the
- maximum magnitude instrumentally recorded in this area (data from https://terremoti.ingv.it/ accessed on  $30^{th}$  May 2025). In recent years, thanks to the installation of the OT local seismic network (see 2.1 for an exhaustive description) the seismic monitoring of GP was strongly enhanced allowing the recording and detection of a high number of micro-earthquakes. As for the above mentioned examples, new insights in the seismic attenuation (Filippucci et al., 2019a; Lucente et al., 2023) and
- et al., 2022) and on seismogenic structures (Miccolis et al., 2021) that could be responsible of both the shallow and deep 70 seismicity in the GP crust were achieved by using the first seismic catalog collected and released so far (Filippucci et al., 2021). This catalog is subject to some limitations as it was collected in real-time mode, at a time when the OT network was

thermo-rheological properties of the upper and lower crust also related to fluid circulation (Filippucci et al., 2019b; Lavecchia

suffering from technical problems in data transmission (see 2.1). With the aim of obtaining a more complete catalog of GP seismicity, in this work we adopted the Complete Automatic Seismic Processor (CASP), (Scafidi et al., 2019), to analyze 10 years of seismic recordings, from 2013 to 2022, in non-real time mode. CASP has been proven to be fast in processing
big amount of 24h seismograms, to be consistent in arrival times picking of P and S phases allowing accurate event location (Spallarossa et al., 2021). The seismic catalog obtained automatically by CASP was manually revised to evaluate the reliability in detection and location. The manual revision of P and S phase arrivals provided two seismic catalogs, by using both linearized (Hypo71) and non-linear (NonLinLoc) algorithms. The quality of the locations was assessed by using the location parameters in a quality factor formula. Magnitudes were also computed by using Di Bona (2016) attenuation law. An analysis of the performance of the seismic network was assessed.

### 2 Gargano seismic network (GSN)

The microseismicity in the Gargano Promontory and surroundings is monitored by the OTRIONS Seismic Network (FDSN code OT, University of Bari "Aldo Moro", 2013), managed by UniBa (Tallarico, 2015), and by the Rete Sismica Nazionale (FDSN code IV, Istituto Nazionale di Geofisica e Vulcanologia (INGV), 2005) managed by INGV. In this paper, we will refer

85 to the Gargano Seismic Network (hereafter GSN) as to a network for the seismic monitoring of the GP area that includes 11 selected stations of the OT network and 10 selected stations of the IV network (Fig. 2) resulting in a very dense network optimized for this study.

### 2.1 OTRIONS seismic network (OT)

In 2013, the OTRIONS (multi-parametric network for the study and monitoring of natural hazards in the OTRanto channel
and IONian Sea) project was funded in the context of the "European Territorial Cooperation Programme Greece-Italy 2007-2013" (INTEREG III) and one of the goals was to deploy a local seismic network around the GP (Tallarico, 2015). The first configuration of the OTRIONS seismic network, refers to years 2013-2014, consisted in 12 seismic stations in the Gargano Promontory: OT01, OT02, OT03, OT04, OT05, OT06, OT07, OT08, OT09, OT10, OT11, OT12. In June 2015, OT01, OT02, OT08 and OT09 were disabled and two new stations installed in the North of the area to provide a better coverage of the
Northern part of the GP (OT13 and OT14 respectively). OT10 was disabled in 2019 due to technical problems. In 2021, two seismic stations were added (OT16 and OT17). In Fig. 2 the actual OT network is shown, where the blue triangles refer to the

active recording stations and black triangles refer to disabled stations.

In order to detect eventual electromagnetic signals related to seismic activity in GP, the station OT04 hosts, since September 2021, a magnetotelluric sensors as described by Ventola et al. (2024). In April 2024, 5 OT stations (OT05, OT11, OT12, OT14,

100 OT16) have been renewed by changing the short-period Lennartz 3D-Lite (1s) seismometers with a broadband sismometer Nanometrics Trillium Compact (20s). The station OT11 was integrated with a Nanometrics Titan accelerometer in the same manhole where the broadband sismometer is housed. Moreover, in the same station has been added a co-located station (OTP1) equipped with a broadband posthole seismometer (Nanometrics T120s-PH3) in a 30 m deep well, providing the simultaneously