# Peer review of "The new seismic catalog of the Gargano area (Southern Italy) after a decade of seismic monitoring by OTRIONS network"

_Earth System Science Data, 2025_

## Referee Comment (RC1)

1. The seismic catalog covers the years 2013-2022. What is the rationale for stopping at 2022? Why not include more recent seismic activity, such as that from 2023-2025? How can users consistently extend the catalog to include more recent data? Is the catalog for this area now consistent with the national one available from INGV?

2. The authors used the CASP tool to detect, pick, and locate events, and they emphasized its advantages, particularly for detecting small events, while also noting the need for manual revision of automatic picks (especially for S-waves). In the last decade machine-learning approaches for event detection, phase picking, and event association have become also increasingly common. Have the authors considered testing such tools and comparing their performance with CASP? For example, many state-of-the-art models are available through *SeisBench* (https://seisbench.readthedocs.io/en/stable/index.html), including models trained on Italian datasets such as INSTANCE, prepared by INGV.

3. Building on the previous observation, machine learning tools are also effective for classification problems, such as distinguishing tectonic events from quarry blasts. Have the authors considered applying one of the tools continuously proposed in the literature to validate their simple approach and confirm the large number of quarry blasts detected in the area? Did the authors check the waveforms and their spectra to confirm the plausibility of the quarry blast records?

4. The authors wrote: *"In this paper, we will refer to the Gargano Seismic Network (hereafter GSN) as to a network for the seismic monitoring of the GP area that includes 11 selected stations of the OT network and 10 selected stations of the IV network (Fig. 2 ) resulting in a very dense network optimized for this study."*. To better understand the suitability of the network geometry with respect to the considered seismicity, it would be helpful to add in Figure 2 the location of the events included in the catalog (Figure 13). This would provide an immediate view of the suitability of the network geometry for monitoring the seismicity of interest. Are the mean statistical location errors in Table 3 too large for an optimized, dense local network? The depth of the identified quarry blasts (Figure 7c) also seems to indicate large uncertainty in depth location, assuming that quarry blasts are shallow.

5. In Table A1, the availability for OT stations MASS, CGL1 and TAR1 is missing.

The authors could explain how to use standard webservices to obtain information about the data availability from 2019. For example, by merging gaps shorter than one day, the availability for station OT07, channel EHE can be obtained through the following web request:
*https://webservices.ingv.it/fdsnws/availability/1/query?network=OT&station=OT07&start=2019-01-01T00:00:00&end=2024-12-31T00:00:00&mergegaps=86400&channel=EHE*

output:

```
**Network Station Location Channel Quality SampleRate Earliest                    Latest**
OT       OT07             EHE     D       100.0      2019-06-13T12:31:58.220000Z 2019-06-29T00:00:00.000000Z
OT       OT07             EHE     D       100.0      2019-07-01T00:50:53.150000Z 2019-07-16T03:50:34.940000Z
OT       OT07             EHE     D       100.0      2019-08-08T14:11:39.030000Z 2019-08-16T02:50:32.160000Z
OT       OT07             EHE     D       100.0      2019-09-03T08:21:06.490000Z 2019-11-07T00:13:14.390000Z
OT       OT07             EHE     D       100.0      2020-11-13T14:15:31.910000Z 2022-07-08T05:28:05.190000Z
OT       OT07             EHE     D       100.0      2022-07-19T11:13:13.650000Z 2024-12-31T00:00:00.000000Z
```

and this would also allow users to check the availability after 2022, and with different tolerances on gap duration.

For users who are not experienced in using FDSN web services, I suggest that the authors also indicate how to use the INGV station web service to obtain information about the OT network. For example:
*https://webservices.ingv.it/fdsnws/station/1/query?level=channel&network=OT&format=text*

```
**Network | Station | location | Channel | Latitude | Longitude | Elevation | Depth | Azimuth | Dip |**
SensorDescription | Scale | ScaleFreq | ScaleUnits | SampleRate | StartTime | EndTime
OT|CGL1||HHE|40.648402|17.517326|303|0|90|0|NANOMETRICS TRILLIUM-40S|1500000000|0.2|m/s|100|2019-05-16T12:03:03|
OT|CGL1||HHN|40.648402|17.517326|303|0|0|0|NANOMETRICS TRILLIUM-40S|1500000000|0.2|m/s|100|2019-05-16T12:03:03|
OT|CGL1||HHZ|40.648402|17.517326|303|0|0|-90|NANOMETRICS TRILLIUM-40S|1500000000|0.2|m/s|100|2019-05-16T12:03:03|
OT|MASS||EHE|40.633|17.144|274|0|90|0|LENNARTZ LE3D-LITE|400|5|m/s|100|2024-11-28T10:55:00|
OT|MASS||EHN|40.633|17.144|274|0|0|0|LENNARTZ LE3D-LITE|400|5|m/s|100|2024-11-28T10:55:00|
OT|MASS||EHZ|40.633|17.144|274|0|0|-90|LENNARTZ LE3D-LITE|400|5|m/s|100|2024-11-28T10:55:00|
OT|MASS||HHE|40.633|17.144|274|0|90|0|NANOMETRICS TRILLIUM-120C|299640000|1|m/s|100|2019-05-16T11:59:46|2024-11-
28T10:55:00
OT|MASS||HHN|40.633|17.144|274|0|0|0|NANOMETRICS TRILLIUM-120C|299640000|1|m/s|100|2019-05-16T11:59:46|2024-11-
28T10:55:00
OT|MASS||HHZ|40.633|17.144|274|0|0|-90|NANOMETRICS TRILLIUM-120C|299640000|1|m/s|100|2019-05-16T11:59:46|2024-11-
28T10:55:00
OT|OT03||EHE|41.712201|15.649727|655|0|90|0|LENNARTZ LE3D-LITE|1677720000|5|m/s|100|2019-05-16T10:46:32|
OT|OT03||EHN|41.712201|15.649727|655|0|0|0|LENNARTZ LE3D-LITE|1677720000|5|m/s|100|2019-05-16T10:46:32|
OT|OT03||EHZ|41.712201|15.649727|655|0|0|-90|LENNARTZ LE3D-LITE|1677720000|5|m/s|100|2019-05-16T10:46:32|
OT|OT04||EHE|41.719584|15.580701|279|0|90|0|LENNARTZ LE3D-LITE|1677720000|5|m/s|100|2019-05-16T10:53:22|
OT|OT04||EHN|41.719584|15.580701|279|0|0|0|LENNARTZ LE3D-LITE|1677720000|5|m/s|100|2019-05-16T10:53:22|
OT|OT04||EHZ|41.719584|15.580701|279|0|0|-90|LENNARTZ LE3D-LITE|1677720000|5|m/s|100|2019-05-16T10:53:22|
```

…

6. The authors computed the local magnitude using Di Bona's (2016) model, which was calibrated for Italy. However, the dataset used to calibrate the local magnitude included very few events and stations from the Gargano area. It would be helpful if the authors shared as an additional asset, the Wood-Anderson amplitudes used to calculate the local magnitude, along with the associated station and event information. This would allow users interested in magnitude to calibrate a local magnitude scale with station corrections specific to the Gargano area, as propagation effects and source parameters (e.g., stress drop) could differ significantly from the average in Di Bona's catalog, particularly for deep events. Furthermore, the Di Bona model was mostly calibrated for Ml>2.8, whereas most of the magnitudes considered in the manuscript are below 2.

---

## Referee Comment (RC2)

**Review of "The new seismic catalog of the Gargano area (Southern Italy) after a decade of seismic monitoring by OTRIONS network" from Ferreri et al.**

The authors present a new catalog obtained with seismic data from the OTRION network in the Gargano area (Italia) over the period 2013 - 2022. Previous catalogs in the region are from the national network, and from OTRION network over the period 2013-2018 (Filippucci et al. 2021) and 2013-2020 (Miccolis et al. 2021). These previous catalogues are from real time processes, suffering of incomplete dataset because of transmission problems. To be able to have a more complete catalog and try to improve the detection of small events, the authors reprocessed the whole period from continuous data with CASP automatic process, and analysed its performance, by comparing the obtained catalog to the manually reviewed version, and previous available catalogs.

**General comments**

The introduction need to be improved to better introduce the work and highlight the contribution of this new catalog. It would be more interesting to focus on processes to build high-quality seismic catalogs and what is already known on the studied area, rather than on a listing of existing networks in South Italia.

More details on CASP must be given, especially on triggering, detection, and picking to be able to appreciate the performance of the process, but also to understand the issues encountered by the authors in their automatic catalog.

The performance of CASP as parametrised in this study does not seem optimal given the significant part of fake events. Are the parametrisation use by the authors for CASP can be still improved? If yes, the authors should discuss this point in the discussion part with some suggestions for improvement. A more general question about the methodology used in this study to build the catalog is: Why did the authors choose CASP which uses rather standard processes, and not more up to date methodologies including deep-learning for automatic picking, new associators, etc?

Some suggestions are made below to improve English, but this is not exhaustive. Careful proofreading should be done to improve the English.

**Specific comments and suggestions**

- p1, line 7: '7 years' instead of '7 year';
- p.1, line 14-16: the structure of the sentence is a bit weird, rephrase;
- p. 2, line 30: 'earthquake' appears twice in the sentence, I suggest 'strong earthquake of November 23, 1999 (Ms=6.9)';
- p. 3, line 60-61: a parallel fault system is mentioned in the text, but not represented in Fig 1b. Add it;
- p. 3, line 61-64: the authors mentioned here the largest earthquakes in the region. It will be very useful in this introduction part to add a seismicity map. It will give an overview of the seismic activity in the area at least before OTRION network (or with the mentioned previous catalog from Filipucci et al. 2021) to better appreciate and highlight the contribution of this network and the catalog obtained in this study. It will also help to locate the mentioned earthquakes;
- p. 4, line 75: I am not sure to understand what the authors mean by 'consistent in arrival times picking'. Consistent compare to what?

- p. 5, line 108 109: I suggest to modify the sentence as follows: 'the network performance was evaluated by the percentage of operating days per year for each station between 2013 and 2022 (tab. A1). It is worth noting the clear improvement in network performance after 2019 compared to the preceding period.';
- p. 5, line 113: 'after 2019' instead of 'after the 2019';
- p. 6, section 2.2 should be shortened: lines 125-127 should be moved to the previous section describing the network, and lines 127 to 133 should be moved in the legend of Figure 3;
- p. 6, line 136: 'optimal' is not the appropriate term regarding the quality of the stations in Figure 3 and A3 to A6, but I agree that half of the stations have overall good performances. Rephrase. Moreover, PPSDs is not suffisant to assess the overall quality of stations in particular for catalogue purpose, as some stations with low noise level (evaluated through PPSD) can lead to overpicking for some reasons and disturb picking process. Thus the number of detections per day (outside active seismic periods) can be a useful additional information;
- p. 9, table 1: it should be moved to the annexe part, no need to be in the main text;
- p. 9, line 164: CASP is mentioned as an *advanced* software for automatic detection/picking/ event location, however looking in detail, it is quite standard procedures, nothing very new compared to nowadays workflow including deep-learning, new associators, etc. Rephrase;
- p. 9, line 170: why having choosing a band-pass filter between 10 and 25 Hz? The lower bound seems to me a bit high. Is there a reason for that?
- p. 10, lines 171-172: give a bit more details on the parameters used for trigger, association and event detection, for instance minimum number of stations/picks, etc. It will help to better understand the performance of the automatic process described later in the paper (significative number of fake events, etc);
- p. 10, lines 175-186: the authors must give a bit more information concerning the method used for picking. All these details would be useful to better understand the analysis of the automatic process performance;
- p. 10, lines 193-194: the authors mentioned the number of seismograms picked manually. It will be also interesting to add information concerning the number of additional picks provided by the manual revision (number of picks missed by the automatic process);
- p. 10, line 197: an example of fake event seen at 3 stations is mentioned. Does it mean that event detection is made considering a minimum number of stations equal to 3? Detection with only 3 stations inevitably leads to false detections without other criteria to discarded them, or/and to badly located events. If it is the case, this could explain some of the critical issues of the automatic process. Specify and comment;
- p. 11, Figure 4: add information about the filtering used for the plot. Use 'recordings' instead of 'registrations' (check other occurrences, and apply same correction). Some picks on the noisiest traces are quite puzzling, as no clear change in the signal is visible. Comment;
- p. 11, lines 206-207: among the 2012 events identified, what is the proportion of fake events and events outside the network badly located inside?
- p. 12 line 229 : explain the choice of depth selection ;
- p. 13, line 238: natural earthquakes should be represented in Fig 7a by blue dots, but there is no blue dot on the corresponding Figure. Add them;
- p. 13, line 242: why quarry blasts have depth ranging between 3 and 7 km? Does it correspond to the manual or the automatic catalog? Why not having relocate them with a fixed depth at the surface when identified as a quarry blast?
- p. 14, Figure 7: add transparency to histogram plots to be able to better see the differences of distribution between natural earthquakes and quarry blasts;

- p. 15, line 254: the value for the standard deviation (2.586), is not consistent with results in Figure 8. Correct it.
- p. 15, line 256-257: the performance of the picker does not seem to be as good as expected. Automatic picks seems to be systematically earlier than the manual ones, and some with significative differences. What is the methodology used for the picking process? As mentioned, this could results in event closer to the network, but also in shallower events. Specify;
- p. 16, line 271: 'assess' instead of 'asses';
- p. 17, lines 286-287: the authors mentioned the fact that rms refers to location quality in time domain, it is not quite exact, it represents the misfit between observed and theoretical traveltimes, it can be affected by the quality of the location in space, and the inconsistency of the true velocity structure compared to the velocity model used. Rephrase. They also mentioned the good values for rms parameters, but looking at table 3, we can noticed that the max value for rms is 23s which is quite high and surprising for a manually revised catalogue! Comment and reprocess the corresponding event;
- Section 5: the first map of the seismicity appears only at the end of this section. It will be interesting to have a map of the seismicity relocated manually with the stations at the beginning of the section to better appreciate the discussions on the catalogue (quality, comparison NLL, hypo71, etc);
- p. 17, lines 300-304: This part is very confusing and not really consistent with Figure 10: delta\_epi is not gaussian, values of standard deviations are very different to values after ± in delta\_epi, delta\_depth and delta\_origin which are very high and not corresponding to figure 10. Correct and rephrase;
- p. 17, line 308: The authors claim that, for earthquake, manual revision is not necessary. But it is needed to remove fake events and quarry blast. Moreover this depends on the purpose, meaning on the use made of the catalog, especially in terms of pick quality. Rephrase
- p. 18, Figure 9 and p.19 line 320: histograms for hypo71 catalog are in red, not in orange. Correct.
- p. 19, line 310: I do not understand the choice made by the authors concerning the initial depths used to refined the depth of locations done with hypo71: depth range is from 10 to 100 km with a step of 10km. Figure 7c clearly shows that most of the events are located above 35km. Comment.
- p. 19, lines 324-326: the difference in errh and errz between event locations done with hypo71 and NLL are surprising. Hypo71 and NLL have quite different uncertainty definitions, in hypo71 uncertainties correspond to one standard deviation, NLL provides a full probability density description of the errors. How errh and errz are computed from NLL output parameters? Check the consistency of errh and errz for this 2 algorithms;
- p. 19, line 327, p. 20, line 328-329: the comments on depth ± errz for events close to the surface, is quite useless. Events close to the surface are always more difficult to locate, and in that sense, NLL often give a better uncertainty estimation, better catching this difficulties. Moreover, when station elevations and topography are included in NLL location, there are less events located at the surface;
- p.20, line 339: in the quality parameter computation (following Michele et al. 2019), why having considering a different weight for the gap? Indeed, azimutal gap has a great influence on the quality of the location, much more than the rms or the number of phases. Comment;
- p. 20, lines 341-345: as the quality parameter *qf* is not exactly the same for NLL and hypo71 as some parameters are not available for hypo71, thus I suspect that *qf* can be slightly different for equivalent quality of location. Thus, as classes are split based on the same value 0/0.25/0.5/0.75/1, the classes for NLL and hypo71 can therefore have slightly different

- characteristic in terms of gap/rms/errh/errz/.... Thus the comparison of percentage of events in the different classes in table 4 may be not quite representative, and misleading. Comment;
- p. 22, line 360: a parenthesis is incorrectly positioned: by Woessner and Wiemer (2005);
- p. 23, lines 365-366: how the authors can claim that the minimal magnitude detectable is lower for this catalog, given that the magnitudes are not exactly computed the same way, and that the difference mentioned is only 0.01 which is well below the typical uncertainties on magnitudes. Rephrase;
- Figure 13: labels of the different plot axes are too small. Enlarge them;
- Figure 14: histogram for hypo71 catalog is in red, not in orange;
- p. 25, line 390: the qf value given for NLL is probably incorrect: 6.3;
- p. 25, lines 394-401: to better appreciate the contribution of this catalog in comparison to the ONT one, seismicity of ONT can be for example plot in the introduction part as already mentioned. A magnitude threshold of -1 is mentioned, this is not consistent with what is mentioned in previous section;
- p. 25, line 400 : reference to figure is not correct : ??
- p. 25, line 411: the authors mentioned the fact that the seismicity follows 2 alignments SW-NE. This can not clearly seen on the figure 13, seismicity is rather diffuse on both side of Mattinata fault;
- p. 25, line 421-422: the authors mentioned a seismic gap, but it sounds more as a region without seismicity. Is there historically seismicity in that region?
- Figure A1 or A7: one of these 2 figures should be put in the main text to better appreciate the discussion on seismicity features.

---

## Author Comment (AC1)

**Answers to comments of Reviewer#2**

December 5, 2025

Reviewer#2:

General comments

The introduction need to be improved to better introduce the work and highlight the contribution of this new catalog. It would be more interesting to focus on processes to build high-quality seismic catalogs and what is already known on the studied area, rather than on a listing of existing networks in South Italia. More details on CASP must be given, especially on triggering, detection, and picking to be able to appreciate the performance of the process, but also to understand the issues encountered by the authors in their automatic catalog. The performance of CASP as parametrised in this study does not seem optimal given the significant part of fake events. Are the parametrisation use by the authors for CASP can be still improved ? If yes, the authors should discuss this point in the discussion part with some suggestions for improvement. A more general question about the methodology used in this study to build the catalog is : Why did the authors choose CASP which uses rather standard processes, and not more up to date methodologies including deep-learning for automatic picking, new associators, etc ? Some suggestions are made below to improve English, but this is not exhaustive. Careful proofreading should be done to improve the English.

Authors:

We rewrote some parts of the introduction section to include processes to build high-quality seismic catalogs taking as an example the paper of Spallarossa et al., 2021 which used CASP software to build high resolution earthquake catalogs for Amatrice (Central Italy) 2016 seismic sequence. We also added in the Introduction section more information about the current knowledge of the Gargano seismotectonics and synthesized the list of existing networks in Southern Italy. In the section "Data Analysis" subsection "CASP automatic list of events - AL" we added more details about the configuration parameters of the CASP modules *Trigger*, *Detect* and *Picker2*. The parameters of the CASP modules was optimized and chosen thorough numerous tests on 100 days of recordings of the GSN network. During these tests we checked the reliability of the filtering and time windows for the STALTA algorithm concluding that the best configuration to maximize the detect is that reported in the manuscript ("An appropriate STA/LTA = 0.8/25 and filtering (band pass filter between

10 Hz and 25 Hz), based on these tests, was selected as best choice.") We also checked the reliability of the Picker2 module by modifying the parameters for S phase detection and reported the best choice in the text ("In this module configuration, the $PostSTeo$ parameter, maximum time interval between the P-wave and S-wave picks, was set to a value determined through several tests, which demonstrated the sensitivity of event detection to this parameter. The optimal value for our study area is $PostSTeo = 5$ s). We also extended the description of the $Detect$ module To help the user that would like to follow the same workflow that we used, in the Data Availability section we released also the configuration files used in CASP modules.

Regarding the optimization of the parameter settings to avoid the detection of false events, we have to do a consideration. Our aim was to enhance the detect and collect a catalog large as much as possible. For this reason we have set the CASP parameters (STA/LTA time windows that are usually (2,80) was set to (0.8,25)) by pushing the detection towards even smaller events, with the disadvantage of inserting a greater number of false events into the catalog, aware that we would have done a manual review of the entire list of events. In order to use CASP without manual revision, an improvement of the setting parameters should be studied and tested. For the same reason, we set 3 phases as minimum for detection increasing the number of false events. See also the response to comment "- p. 10, line 197" We added this comment in the discussion section. The choice of CASP to build the seismic catalog of GP does not exclude the use of machine learning methods to do the same task. As already answered to Reviewer#1: The CASP tool is successfully implemented for the seismic monitoring of Northwestern Italy. So we explored the possibility to implement this code for the seismic monitoring of the Gargano Promontory. The CASP license was acquired thank to a collaboration between the University of Bari Aldo Moro and the INGV. The code is nowadays operative. The CASP code gave us the possibility to analyze ten years of seismic recordings in offline mode with the results illustrated in the manuscript. We conclude that the CASP software is able to detect a large number of small magnitude events, the S-wave picking suffers of some uncertainties that do not affect the reliability of the final location, manual revision is necessary to exclude from the catalog the non-earthquake events. Overall the CASP software was useful to build the released catalog. Manual review of the catalog obtained with CASP allowed us to collect a dataset of highly reliable earthquakes, as they were repicked. This dataset, for which we have released a bulletin in the Data Availability section, will allow us to test the effectiveness of machine learning techniques in automatic picking and building automatic catalogs. In fact, the machine learning detection (PhaseNet, Zhu et a., 2019) and association techniques (GAMMA, Zhu et al., 2021) can be applied to the same dataset, leading to a greater number of detected events. The results of this study and the well located earthquakes of the released catalogs can be used to test the effectiveness of machine learning techniques in automatic picking when building automatic catalogs. We modified the discussion section to account for this comment.

Reviewer#2:
Specific comments and suggestions
− p1, line 7: '7 years' instead of '7 year';
− p.1, line 14-16 : the structure of the sentence is a bit weird, rephrase ;
− p. 2, line 30 : 'earthquake' appears twice in the sentence, I suggest 'strong earthquake of November 23, 1999 (Ms=6.9)' ;
− p. 3, line 60-61 : a parallel fault system is mentioned in the text, but not represented in Fig 1b. Add it ;

Authors:
We modifies the text and changed the Figure 1 to include the parallel fault system (black line) mentioned in the text.

Reviewer#2:
- p. 3, line 61-64 : the authors mentioned here the largest earthquakes in the region. It will be very useful in this introduction part to add a seismicity map. It will give an overview of the seismic activity in the area at least before OTRIONS network (or with the mentioned previous catalog from Filippucci et al. 2021) to better appreciate and highlight the contribution of this network and the catalog obtained in this study. It will also help to locate the mentioned earthquakes ;

Authors:
We added on the map the location of the two earthquakes mentioned in the text.

Reviewer#2:
- p. 4, line 75 : I am not sure to understand what the authors mean by 'consistent in arrival times picking'. Consistent compare to what ?

Authors:
The term "consistent" refers to the reliability of CASP in the picking procedure. We modified the sentence for more clarity.

Reviewer#2:

- p. 5, line 108 - 109 : I suggest to modify the sentence as follows : 'the network performance was evaluated by the percentage of operating days per year for each station between 2013 and 2022 (tab. A1). It is worth noting the clear improvement in network performance after 2019 compared to the preceding period.' ;
- p. 5, line 113 : 'after 2019' instead of 'after the 2019' ;
- p. 6, section 2.2 should be shortened : lines 125-127 should be moved to the previous section describing the network, and lines 127 to 133 should be moved in the legend of Figure 3 ;

Authors:

We modified the text to account for the Reviewer#2 suggestions.

Reviewer#2

- p. 6, line 136 : 'optimal' is not the appropriate term regarding the quality of the stations in Figure 3 and A3 to A6, but I agree that half of the stations have overall good performances. Rephrase. Moreover, PPSDs is not suffisant to assess the overall quality of stations in particular for catalogue purpose, as some stations with low noise level (evaluated through PPSD) can lead to overpicking for some reasons and disturb picking process. Thus the number of detections per day (outside active seismic periods) can be a useful additional information;

Authors:

The text was modified and the term "optimal" was substituted with "overall good". The detection was evaluated by counting the number of P and S phases picking for each station every three months for the decade from 2013 to 2022. The results are organized in the histograms in the following figure.

[Figure]

Figure 1: Histograms of number of automatic picking, used for EQ catalogs, for each station of the OT network, from 2013 to 2022 with bins of three months. Superimposed the total number of pickings

From the comparison of the PPSD and the histograms of automatic pickings, some considerations can be done. In the frequency range $[10, 25]$ Hz of the STALTA algorithm of CASP used for detection, the stations with the lowest level of noise (OT01, OT03, OT04, OT05, OT06, OT07, OT08, OT09) and those with an intermediate noise level (OT12, OT14 and OT16) are characterized by the highest number of pickings in the period of operation. The stations OT05 and OT06 were used as an unique station given their closeness. The stations with a high noise level (OT02, OT10, OT11, OT13 and OT17) correspond to the stations that contributed less to the detection. The correlation between the level of the noise and the number of automatic pickings is worth to be discussed and we added it in the manuscript in section Supplementary material.

Reviewer#2

- p. 9, table 1 : it should be moved to the annexe part, no need to be in the main text ;
- p. 9, line 164 : CASP is mentioned as an advanced software for automatic detection/picking/ event location, however looking in detail, it is quite standard procedures, nothing very new compared to nowadays workflow including deep-learning, new associators, etc. Rephrase ;
- p. 9, line 170 : why having choosing a band-pass filter between 10 and 25 Hz ? The lower bound seems to me a bit high. Is there a reason for that ?

Authors:

Table 1 was moved to the Supplementary material in Section A1.
We eliminated the qualitative adjective "advanced".
Regarding the band pass filter for STALTA we referred to the work of Spallarossa et al. (2021) who used a band-pass filter between [10,30] Hz and adapted it to our data after a training tests over 100 days of registration. We added this explanation in the text.

Reviewer#2

- p. 10, lines 171-172 : give a bit more details on the parameters used for trigger, association and event detection, for instance minimum number of stations/picks, etc. It will help to better understand the performance of the automatic process described later in the paper (significative number of fake events, etc) ;
- p. 10, lines 175-186 : the authors must give a bit more information concerning the method used for picking. All these details would be useful to better understand the analysis of the automatic process performance ;
- p. 10, lines 193-194: the authors mentioned the number of seismograms picked manually. It will be also interesting to add information concerning the number of additional picks provided by the manual revision (number of picks missed by the automatic process) ;
- p. 10, line 197 : an example of fake event seen at 3 stations is mentioned. Does it mean that event detection is made considering a minimum number of stations equal to 3 ? Detection with only 3 stations inevitably leads to false detections without other criteria to discarded them, or/and to badly located events. If it is the case, this could explain some of the critical issues of the automatic process. Specify and comment ;
- p. 11, Figure 4 : add information about the filtering used for the plot. Use 'recordings' instead of 'registrations' (check other occurrences, and apply same correction). Some picks on the noisiest traces are quite puzzling, as no clear change in the signal is visible. Comment ;

- p. 11, lines 206-207 : among the 2012 events identified, what is the proportion of fake events and events outside the network badly located inside ?

Authors:

In the description of the modules of the CASP software, *Trigger*, *Detect* and *Picker2*, we added some more details and we also added in the online repository (Ferreri et al., 2025) the configuration files used as input files for the CASP modules.
To evaluate the usefulness of the manual picking procedure in terms of number of pickings, we counted both the automatic and the manual ones only for the earthquakes (4098 EQ) of the catalogs. In the following figure do not appear all the automatic pickings of the CASP software that refer to the automatic list of events (7162 AL).

[Figure]

Figure 2: Histogram of the number of automatic (purple) and manual (green) pickings for the earthquake catalog EQ carried out on all OT stations

The histogram reveals that the manual procedure, slightly but systematically for all the OT stations, increases the number of pickings. We added this new figure to the manuscript in the supplementary material.
The minimum number of trigger to detection is specified in the configuration file and it is equal to 3. The example in the figure is a false event detected at 13 stations. We modified the figure to solve the misunderstanding.
To understand the percentage of false and bad located events respect to the events automatically detected, we proceeded to label the events in the category FB and created two new categories: the false events (FE) and the bad located earthquakes (BL). This procedure gave us the possibility to control again our data and correct the number of events in the automatic list of events (AL).

In the following figure (Fig. 3), the number of the events discarded as false (FE) and bad located (BL) is plotted versus the number of phases used for detection. This result indicates that the number of discarded events decreases with the number of phases used for detection and it can be used as an indication to how the CASP tool can work alone without supervision.

We added this figure and this discussion in the manuscript. See also the response to general comment.

[Figure]

Figure 3: Curves of BL and FE (gray and black respectively) plotted versus the number of phases used by CASP for the detection.

Reviewer#2

- p. 12 line 229 : explain the choice of depth selection ;
- p. 13, line 238 : natural earthquakes should be represented in Fig 7a by blue dots, but there is no blue dot on the corresponding Figure. Add them ;
- p. 13, line 242 : why quarry blasts have depth ranging between 3 and 7 km ? Does it correspond to the manual or the automatic catalog ? Why not having relocate them with a fixed depth at the surface when identified as a quarry blast ?
- p. 14, Figure 7 : add transparency to histogram plots to be able to better see the differences of distribution between natural earthquakes and quarry blasts ;

Authors:

Working on the quarry blasts events we observed that some of them automatically located could have foci depth greater than 5 km. For this reason, we extended the depth range in which to search for automatically detected quarry explosions. We modified the phrase.

We modified the figure to account for the request and added EQ in the map. The quarry blasts plotted in map are located by the automatic procedure of the CASP tool. No re-picking neither relocation was carried out on quarry blasts, but they were only discarded from the final catalogs. We agree that the catalog of quarry blasts deserves a dedicated study and this was behind the scope of this paper. We clarified this motivation in the text

The daytime and depth histograms of quarry blasts is a stacked histogram, thus the yellow bar indicating the number of blasts is over the blue bar indicating the number of earthquakes. We clarified in the caption

Reviewer#2

- p. 15, line 254 : the value for the standard deviation (2.586), is not consistent with results in Figure 8. Correct it.
- p. 15, line 256-257 : the performance of the picker does not seem to be as good as expected. Automatic picks seems to be systematically earlier than the manual ones, and some with significative differences. What is the methodology used for the picking process ? As mentioned, this could results in event closer to the network, but also in shallower events. Specify ;
- p. 16, line 271 : 'assess' instead of 'asses' ;

Authors:

We corrected the value of the standard deviation of S picks $\sigma_{\Delta T_S} = 0.654$ s
The automatic picks are later than the manual ones slightly systematically. We already evidenced this behavior in the manuscript. The methodology for automatic picking is already explained in the section "CASP automatic list of events - AL". It is worth to note that nevertheless this discrepancy in the time picks, the effect in the earthquake location shown in Figure 11 (now Figure 12) is negligible and inside the intrinsic error bar of location. All these consideration are already present in the section "NLL-catalog of EQ".

Reviewer#2

p. 17, lines 286-287 : the authors mentioned the fact that rms refers to location quality in time domain, it is not quite exact, it represents the misfit between observed and theoretical travel times, it can be affected by the quality of the location in space, and the inconsistency of the true velocity structure compared to the velocity model used. Rephrase. They also mentioned the good values for rms parameters, but looking at table 3, we can noticed that the max value for rms is 23s which is quite high and surprising for a manually revised catalogue ! Comment and reprocess the corresponding event ;
- p. 17, line 308 : The authors claim that, for earthquake, manual revision is not

necessary. But it is needed to remove fake events and quarry blast. Moreover this depends on the purpose, meaning on the use made of the catalog, especially in terms of pick quality. Rephrase;

Authors:

We corrected the misunderstanding concerning the time domain and rephrased. Regarding the outliers in the manual EQ catalogs, the Table 2 clearly demonstrates that the outliers, obviously included in the catalog, since human errors in manual reviews are unavoidable, do not affect the quality of the final catalogs. In fact the mean, the median and the mode values of the quality parameters in Table 2 are of really good quality. It is important to observe that the statistics in Tab.2 ($95^{th}$ and $5^{th}$) are necessary to compute the quality $q_f$ of the earthquake as described in the formula in Eq. (4). Only for the estimator number of phases nphs we used the maximum value and the 5th percentile as minimum value. The maximum values of rms, erh and erz refer to events that are discarded after the application of the quality formula, as explained in the section 5.3. This demonstrate that the $q_f$ parameter is well formulated and useful to automatically clean the catalog from low-quality locations. We modified the table 2 and eliminated the rows of 50th percentile and 90th percentile since they are not used in the formula and also eliminated the row indicating the maximum and the minimum of the uncertainty estimator since they are not used in the quality formula, with the exception of the maximum of $nphs$. We also added a conclusive table of statistics which refer only to the final released catalogs, after the quality selection.
We rephrased as following: "Once earthquake-type events have been identified and selected, their location in terms of P and S phase recognition is acceptable even without manual review, as shown in Fig. 12. If the effort to catalog the event type were automated, for example with artificial intelligence techniques trained on the specific dataset, rather than manual, the CASP software could operate without supervision. However, if high-quality picking is required, manual review of time arrivals is still necessary, as shown in Fig. 10."

Reviewer#2

- p. 18, Figure 9 and p.19 line 320: histograms for hypo71 catalog are in red, not in orange. Correct.
p. 19, line 310 : I do not understand the choice made by the authors concerning the initial depths used to refined the depth of locations done with hypo71 : depth range is from 10 to 100 km with a step of 10km. Figure 7c clearly shows that most of the events are located above 35km. Comment.

Authors:

We modified the caption for a better explanation: "Overlapped histograms of quality location estimators for the H71-catalog (orange+red areas) and NLL-

catalog (blue+red areas) . The areas in red color should be considered as common parts for both the H71 and NLL catalogs.".

The choice of the number of initial values of hypocentral depth to start the $H71$ inversion is arbitrary and we decided to use 10 different depths in continuity with Miccolis et al. (2021) even if we could stop the investigation at a depth of 50 km. The result does not change.

Reviewer#2

- p. 19, lines 324-326 : the difference in errh and errz between event locations done with hypo71 and NLL are surprising. Hypo71 and NLL have quite different uncertainty definitions, in hypo71 uncertainties correspond to one standard deviation, NLL provides a full probability density description of the errors. How errh and errz are computed from NLL output parameters ? Check the consistency of errh and errz for this 2 algorithms ; p. 19, line 327, p. 20, line 328-329 : the comments on depth $\pm$ errz for events close to the surface, is quite useless. Events close to the surface are always more difficult to locate, and in that sense, NLL often give a better uncertainty estimation, better catching this difficulties. Moreover, when station elevations and topography are included in NLL location, there are less events located at the surface ;

Authors:

The software NLL gives as output files also one file written in h71 format. The conversion from the uncertainty provided by NLL in h71 format is computed and released by Lomax et al. (2000 and 2014) and the conversion is explained in the webpage of the NLL code (http://alomax.free.fr/nlloc/). We added this explanation in the text.

We agree with the Reviewer#2 regarding the shallow earthquakes and in fact the those with vertical error bar such as to bring the focus off the surface were automatically discarded from the final catalog thanks to the application of the quality formula. In this work the quality formula reveals a very important tool for automatically select the best quality earthquakes.

Reviewer#2

- p.20, line 339 : in the quality parameter computation (following Michele et al. 2019), why having considering a different weight for the gap ? Indeed, azimutal gap has a great influence on the quality of the location, much more than the rms or the number of phases. Comment ;

- p. 20, lines 341-345 : as the quality parameter qf is not exactly the same for NLL and hypo71 as some parameters are not available for hypo71, thus I suspect that qf can be slightly different for equivalent quality of location. Thus, as classes are split based on the same value 0/0.25/0.5/0.75/1, the classes for NLL and hypo71 can therefore have slightly different characteristic in terms of gap/rms/errh/errz/.... Thus the comparison of percentage of events in the

different classes in table 4 may be not quite representative, and misleading. Comment ;

Authors:

We agree with Reviewer#2 that the azimuthal gap has a very important role in the quality of the earthquakes but the Gargano Promontory for its geographical characteristics can be assimilated to an offshore area and the earthquakes occurred in the Gargano Promontory should be treated as offshore earthquakes. For this reason we weighted the azimuthal gap less than other quality parameters as proposed by Latorre et al., 2023 to prevent the quality of the entire catalog from being brought down by the quality of the gap. We added this explanation in the manuscript.

As the Reviewer#2 comments, the differences in the location quality between NLL and H71 codes are not only due the parameters locdist and rpdf, which are present in NLL output and not in H71 output, but also in the values of erh and erz, whch are present in the putput of both NLL and H71, as it can be seen in Figure 11. Since the quality as retrieved from the quality formula, depends on the values of the error parameters, the $qf$ value will never be the same for a given earthquake located with both NLL and H71 codes. However, the two final quality results for the entire catalog do not differ much and lead to the same conclusion, namely that most earthquakes belong to the first three quality classes. The result of table 4 are therefore representative of the quality of locations, as shown by the location errors in Fig. 11.

Reviewer#2

- p. 23, lines 365-366 : how the authors can claim that the minimal magnitude detectable is lower for this catalog, given that the magnitudes are not exactly computed the same way, and that the difference mentioned is only 0.01 which is well below the typical uncertainties on magnitudes. Rephrase ;

Authors:

The minimum detectable magnitude moves from -0.1 of the preceding catalog (Filippucci et al., 2021) to -1 of this one. So the difference is of 0.9 degrees of magnitude not of 0.01, greater than uncertainties on magnitude. To avoid misunderstanding we substituted -0.99 with -1.

Reviewer#2

- Figure 13 : labels of the different plot axes are too small. Enlarge them ;
- Figure 14 : histogram for hypo71 catalog is in red, not in orange ;

Authors:

We modified the labels in Figure 13 (now Figure 15)
We clarified the meaning of the colors in the caption of the Figure 14.

Reviewer#2

- p. 25, line 390 : the qf value given for NLL is probably incorrect : 6.3
;
p. 25, lines 394-401 : to better appreciate the contribution of this catalog in
comparison to the ONT one, seismicity of ONT can be for example plot in the
introduction part as already mentioned. A magnitude threshold of -1 is mentioned, this is not consistent with what is mentioned in previous section ;

Authors:

Regarding the group of earthquakes that NLL locates at shallow depths, with
such a vertical error that the events could be above the Earth's surface, these
earthquakes are 418 and have an average depth of $1.95 \pm 6.88$ km (Fig. 13)
and an average quality qf = 6.3 in the NLL catalog and were discarded from
the final NLL-catalog. This example demonstrates that the qf parameter in Eq.
4 proposed in this paper is well formulated and qf ¿ 1 is a reliable criterion to
discard bad quality events. These 418 discarded from the NLL-catalog, when
located with H71 result at an average depth of $14.56 \pm 4.03$ km and have an
average qf = 0.57, so they can be included in the H71-catalog. This explains
why, in Tab. 3, the number of earthquakes in each quality class may differ between the NLL and H71 locations.
The lower magnitude threshold calculated with our catalog is exactly -1 as reported in the text (since $-0.99 \simeq -1$)
We decided not to show the seismicity of the ONT catalog in the figures to
avoid confusion with our catalog. For and indication of the number of detected
events, in Figure 4A of the Supplementary material we plotted the cumulative
number of earthquakes of the ONT which shows that our catalog includes more
than twice the number of earthquakes respect to the ONT in the same region.

Reviewer#2

- p. 25, line 400 : reference to figure is not correct : ??
- p. 25, line 411 : the authors mentioned the fact that the seismicity follows
2 alignments SW-NE. This can not clearly seen on the figure 13, seismicity is
rather diffuse on both side of Mattinata fault ;
- p. 25, line 421-422 : the authors mentioned a seismic gap, but it sounds more
as a region without seismicity. Is there historically seismicity in that region ?
- Figure A1 or A7 : one of these 2 figures should be put in the main text to
better appreciate the discussion on seismicity features.

Authors:

We corrected the reference.

In Figure 13 (Figure 15 in the revised manuscript) two clusters can be observed that are elongated in SW-NE direction separated by an area elongated in the same Sw-NE direction characterized by absence of seismicity and located one at the ortern part of GP and the other at the southern part of GP crossing the MF fault. These cluster are clearly visible both with the H71-catalog and with the NLL catalog. We modified the text for more clarity.

The absence of seismicity in NE part of the Gargano Promontory is also documented from the historical seismicity as showed by Del Gaudio et al. (2007) in Fig. 3. We rephrased.

To better appreciate the discussion and following the suggestion of the Reviewer#1, we moved Fig. A1 in the discussion section (Fig.17 in the revised paper).

References:

Del Gaudio, V., Pierri, P., Frepoli, A., Calcagnile, G., Venisti, N. and Cimini, G. B. (2007). A critical revision of the seismicity of Northern Apulia (Adriatic microplate—Southern Italy) and implicationsfor the identification of seismogenic structures. Tectonophysics, 436(1-4), 9-35.https://doi.org/10.1016/j.tecto.2007.02.013

Ferreri, Andrea Pio; Romeo, Annalisa; Giannuzzi, Rossella; Ninivaggi, Teresa; Filippucci, Marilena; Cecere, Gianpaolo; Falco, Luigi; Michele, Maddalena; Selvaggi, Giulio; Tallarico, Andrea (2025), "The new seismic catalog of the Gargano area (Southern Italy) after a decade of seismic monitoring by OTRIONS network", Mendeley Data, V4, https://doi.org/10.17632/nhfvx7ysxw.4

Filippucci, M., Miccolis, S., Castagnozzi, A., Cecere, G., de Lorenzo, S., Donvito, G., Falco, L., Michele, M., Nicotri, S., Romeo, A., Selvaggi, G., Tallarico, A. (2021). Seismicity of the Gargano promontory (Southern Italy) after 7 years of local seismic network operation: Data release of waveforms from 2013 to 2018. Data in Brief, 35, 106783. https://doi.org/10.1016/j.dib.2021.106783

Latorre, D., Di Stefano, R., Castello, B., Michele, M., and Chiaraluce, L.: An updated view of the Italian seismicity from probabilistic location in 3D velocity models: The 1981–2018 Italian catalog of absolute earthquake locations (CLASS), Tectonophysics, 846, 229 664, https://doi.org/https://doi.org/10.1016/j.tecto.2022.229664, 2023.

Lomax A., Virieux J., Volant P., Berge-Thierry C. (2000) Probabilistic Earthquake Location in 3D and Layered Models. In: Thurber C.H., Rabinowitz N. (eds) Advances in Seismic Event Location. Modern Approaches in Geophysics, vol 18. Springer, Dordrecht. https://doi.org/10.1007/978-94-015-9536-0$_$5

Lomax A., Michelini A., Curtis A. (2014) Earthquake Location, Direct, Global-Search Methods. In: Meyers R. (eds) Encyclopedia of Complexity and Systems Science. Springer, New York, NY. `https://doi.org/10.1007/978-3-642-27737-5$_$150-2`

Miccolis, S., Filippucci, M., De Lorenzo, S., Frepoli, A., Pierri, P., Tallarico, A. (2021). Seismogenic structure orientation and stress field of the Gargano Promontory (southern Italy) from microseismicity analysis. Frontiers in Earth Science, 9, 589332.`https://www.frontiersin.org/journals/earth-science/articles/10.3389/feart.2021.589332`

Spallarossa, D., Cattaneo, M., Scafidi, D., Michele, M., Chiaraluce, L., Segou, M., Main, I. G. (2021). An automatically generated high-resolution earthquake catalogue for the 2016–2017 Central Italy seismic sequence, including P and S phase arrival times. Geophysical Journal International, 225(1), 555-571.`https://doi.org/10.1093/gji/ggaa604`

Zhu, W. and Beroza, G. C. (2018). Phasenet: a deep-neural-network-based seismic arrival-time picking method. Geophysical Journal International, 216(1):261–273. `https://doi.org/10.1093/gji/ggy423`

Zhu, W., McBrearty, I. W., Mousavi, S. M., Ellsworth, W. L., and Beroza, G. C. (2022). Earthquake phase association using a bayesian gaussian mixtu-re model. Journal of Geophysical Research: Solid Earth, 127(5):e2021JB023249. `https://doi.org/10.1029/2021JB023249`

---

## Author Comment (AC2)

**Answers to comments of Reviewer#1**

December 5, 2025

We wish to thank Dino Bindi for his helpful comments which gave us the possibility to rewrote some parts of the manuscript that weren't so clear. Hereafter, the responses to his comments.

1.
Reviewer #1:
The seismic catalog covers the years 2013-2022. What is the rationale for stopping at 2022? Why not include more recent seismic activity, such as that from 2023-2025? How can users consistently extend the catalog to include more recent data? Is the catalog for this area now consistent with the national one available from INGV?

Authors:
As illustrated in the manuscript, the OT network works since 2013 and during the 2019 changed the data transmission protocol and recordings are available real time on the web services of EIDA node managed by INGV. So, being data starting from the second half of 2019 already available, our main interest was to enrich the knowledge of the seismicity rates in the period in which data are not available online on the EIDA web service (so for the period from 2013 to the first half of 2019. We decided to extend the analysis up to the end of 2022 to allow a comparison of the detect of the software CASP (Scafidi et al., 2019) with that of the ONT. The analysis of ten years of seismograms was extremely time-consuming so when we stopped the picking phases processing we dedicated to the analysis and evaluation of the results for that decade. The comparison of our catalog with the ONT catalog is then possible only from 2019 to 2022. Being the CASP software set ad hoc for the microseismicity detection, our catalog present a greater number of events of lower magnitude than that of the ONT catalog (2640 in our catalog respect to 1463 of ONT). Among the 1463 of the ONT catalog, there are about 300 earthquakes that CASP did not detect, probably for the different settings in the STA/LTA algorithm. We added this comment in the manuscript, in the discussion section. To extend the analysis to the present, the users may download the waveforms from the INGV webservice of EIDA and should process them with the CASP software (having previously acquired the license) to continue with the same workflow or use some other open source detection codes as SeisComP3.

2.
Reviewer #1:
The authors used the CASP tool to detect, pick, and locate events, and they emphasized its advantages, particularly for detecting small events, while also noting the need for manual revision of automatic picks (especially for S-waves). In the last decade machine-learning approaches for event detection, phase picking, and event association have become also increasingly common. Have the authors considered testing such tools and comparing their performance with CASP? For example, many state-of-the-art models are available through SeisBench (https://seisbench.readthedocs.io/en/stable/index.html), including models trained on Italian datasets such as INSTANCE, prepared by INGV.

Authors:
The CASP tool is successfully implemented for the seismic monitoring of Northwestern Italy. So we explored the possibility to implement this code for the seismic monitoring of the Gargano Promontory. The CASP license was acquired thank to a collaboration between the University of Bari Aldo Moro and the INGV. The code is nowadays operative. The CASP code gave us the possibility to analyze ten years of seismic recordings in offline mode with the results illustrated in the manuscript. We conclude that the CASP software is able to detect a large number of small magnitude events, the S-wave picking suffers of some uncertainties that do not affect the reliability of the final location, manual revision is necessary to exclude from the catalog the non-earthquake events. Overall the CASP software was useful to build the released catalog. Manual review of the catalog obtained with CASP allowed us to collect a dataset of highly reliable earthquakes, as they were re-picked. This dataset, for which we have released a bulletin in the Data Availability section, will allow us to test the effectiveness of machine learning techniques in automatic picking and building automatic catalogs. In fact, the machine learning detection (PhaseNet, Zhu et a., 2019) and association techniques (GAMMA, Zhu et al., 2021) can be applied to the same dataset, leading to a greater number of detected events. The results of this study and the well located earthquakes of the released catalogs can be used to test the effectiveness of machine learning techniques in automatic picking when building automatic catalogs. We modified the discussion section to account for this comment.

3.
Reviewer #1: Building on the previous observation, machine learning tools are also effective for classification problems, such as distinguishing tectonic events from quarry blasts. Have the authors considered applying one of the tools continuously proposed in the literature to validate their simple approach and confirm the large number of quarry blasts detected in the area? Did the authors check the waveforms and their spectra to confirm the plausibility of the quarry blast records?

Authors:

We partially agree about the effectiveness of the machine learning tools to avoid human costs. Our experience, which will be subject of a subsequent dedicated work, indicates that visual revision is still necessary to build a reliable catalog of earthquakes. The machine learning based tool for binary classification (quarry blasts and earthquakes) should work on a catalog where all the false events and the regional events located as local earthquakes are already eliminated from the list. In our case, the catalog includes 4 types of events (earthquakes, quarries, false and bad located) and the binary classificator cannot be implemented, at this stage.

The recognition of a quarry explosion was carried out, as described in the manuscript, on a subset of events, located in the vicinity of known mining quarries, which occurred during working hours on working days, by observing the waveforms of the closest station, where the arrival of the S-wave is absent, and neglecting waveforms of the more distant ones, where a surface wave can be confused with the S-wave arrival. We rewrote to clarify the procedure.

4.

Reviewer #1:

The authors wrote: "In this paper, we will refer to the Gargano Seismic Network (hereafter GSN) as to a network for the seismic monitoring of the GP area that includes 11 selected stations of the OT network and 10 selected stations of the IV network (Fig. 2 ) resulting in a very dense network optimized for this study.". To better understand the suitability of the network geometry with respect to the considered seismicity, it would be helpful to add in Figure 2 the location of the events included in the catalog (Figure 13). This would provide an immediate view of the suitability of the network geometry for monitoring the seismicity of interest. Are the mean statistical location errors in Table 3 too large for an optimized, dense local network? The depth of the identified quarry blasts (Figure 7c) also seems to indicate large uncertainty in depth location, assuming that quarry blasts are shallow.

Authors:

We have already discussed in the manuscript (in the discussion section) the seismicity distribution across the years (Figures are in section Supplementary material) with the different three stations configuration: from April 2013 to June 2015, from July 2015 to March 2021 and from April 2021 to December 2022. This was aimed to better understand how seismicity shifts based on the variation in stations configuration. Regarding location errors, we need to do two considerations. First, the velocity model used for location (de Lorenzo et al. 2017) was developed with the OT network geometry of the early years (2013-2015) when some stations (OT13, OT14, OT16, OT17, TREM) weren't installed yet and some others were subsequently removed (OT01, OT02, OT08, OT09, OT10). We think that a revision of the velocity model is necessary, maybe it is slightly fast and this could be the reason why blasts are located deeper in the crust than they should. Second, the seismicity of the GP is very deep in the crust, at depth between 15 and 25 km, and it is of very low energy, for the

majority of the earthquakes we have $0.1 < M_L < 3$, so the takeoff angle of the most of the recording stations is small and the recordings are few for each event. This could explain why we have horizontal and vertical errors that are typical of regional events. We added these considerations in the discussion section.

5.
Reviewer:
In Table A1, the availability for OT stations MASS, CGL1 and TAR1 is missing.
The authors could explain how to use standard webservices to obtain information about the data availability from 2019.

Authors:
In table A1, we refer only to the Gargano Seismic Network (GSN), which include OT stations and IV stations installed in the Gargano area. Stations MASS, CGL1 and TAR1 are OT stations installed in Massafra (TA), Ceglie Messapica (BR) and Taranto (TA), respectively.
We have included information about INGV web services and their use, adding the following sentence in the manuscript in the data availability section: The availability of the OT stations after 2019 can be downloaded through the standard FDSN web services (https://www.fdsn.org/webservices/), using the INGV webservice. For example, the availability from 2019 to 2014 for station OT07, channel EHE, can be obtained through the following web request:
```
https://webservices.ingv.it/fdsnws/availability/1/query?network=
OT&station=OT07&start=2019-01-01T00:00:00&end=2024-12-31T00:00:
00&mergegaps=86400&channel=EHE
```
This request shows the availability of the channel, ignoring the gaps smaller than 24h (mergegaps=86400 s). To obtain information on seismic station and instrumentation, one possible query can be:
```
https://webservices.ingv.it/fdsnws/station/1/query?level=channel
&network=OT&format=text
```
which returns the geographical coordinates and instrument details of the seismic stations belonging to the network OT. Some other information and examples could be found at the link:
```
https://www.orfeus-eu.org/data/eida/nodes/INGV/
```
6.

Reviewer #1:
The authors computed the local magnitude using Di Bona's (2016) model, which was calibrated for Italy. However, the dataset used to calibrate the local magnitude included very few events and stations from the Gargano area. It would be helpful if the authors shared as an additional asset, the Wood-Anderson amplitudes used to calculate the local magnitude, along with the associated station and event information. This would allow users interested in magnitude to calibrate a local magnitude scale with station corrections specific to the Gargano area, as propagation effects and source parameters (e.g., stress drop) could differ

significantly from the average in Di Bona's catalog, particularly for deep events. Furthermore, the Di Bona model was mostly calibrated for Ml>2.8, whereas most of the magnitudes considered in the manuscript are below 2.

Authors:
We used the Di Bona (2016) local magnitude scale since it is derived for the Italian region and it is based on recent seismicity. Di Bona (2016) attenuation law analyzed the local magnitude bias with the Hutton and Boore (1987), used in the routine magnitude computation in the ONT at INGV, and found that for the Italian region Hutton and Boore (1987) would lead to magnitude overestimation at short-range stations and underestimation farther than 100 km. The magnitude bias varies with distance and increases in absolute value up to $\sim 0.55$ from 100 to 10 km and up to $\sim 0.35$ from 100 to 600 km. Moreover, the effect of the attenuation function of Di Bona (2016) highlights that using HB87 leads to overestimation of the lowest magnitudes and underestimation of the highest magnitudes, with a bias within about 0.2 within a magnitude range from 2.8 to 5.5. In our seismic catalog we have earthquakes both of low magnitude and at short distance so the effect of overestimation of the HB87 attenuation law respect to DB16 could be enhanced. This effect can be observed when computing the differences between the $ML_{DB16}$, used in this work, and the $ML_{HB87}$, used by the ONT. Results in $L_1$ norm of $ML_{DB16} - ML_{HB87}$ are shown in the following histogram (Fig. 1 and in Fig. 2) and indicate that, as expected, the HB87 attenuation law tends to overestimate the local magnitude respect to the DB16 attenuation law. The effect of the overestimation is, as expected and predicted by DB16, increases moving toward very small magnitudes and the HB87 does not compute $M_L < 0$. The bias decreases moving toward greater value of magnitude where the two formulations agree. For $M_L > 2.8$ the bias is less than 0.2 as also observed by DB16. Results indicate that it would be useful to calibrate a magnitude scale for the microseismicity of the Gargano area. We added this discussion in the Section 5.4. Regarding amplitude data, we can share the WA amplitudes of the earthquakes in the released catalogs to anyone who requests them. We added this sentence in the Data availability section.

[Figure]

Figure 1: Histogram of magnitude differences for CASP earthquakes by comparing the magnitude estimates computed using the DB16 attenuation law with those obtained using the HB87 attenuation law.

[Figure]

Figure 2: Correlation between CASP magnitude values computed with DB16 attenuation law (vertical axis) and HB87 attenuation law (horizontal axis).

References:

de Lorenzo, S., Michele, M., Emolo, A., & Tallarico, A. (2017). A 1D P-wave velocity model of the Gargano promontory (south-eastern Italy). Journal of Seismology, 21(4), 909-919.

Di Bona, M. (2016). A local magnitude scale for crustal earthquakes in Italy. Bulletin of the Seismological Society of America, 106(1), 242-258.

Hutton, L. K., & Boore, D. M. (1987). The ML scale in southern California. Bulletin of the Seismological Society of America, 77(6), 2074-2094.

Scafidi, D., Spallarossa, D., Ferretti, G., Barani, S., Castello, B., & Margheriti, L. (2019). A complete automatic procedure to compile reliable seismic catalogs and travel-time and strong-motion parameters datasets. Seismological Research Letters, 90(3), 1308-1317.

Zhu, W., & Beroza, G. C. (2019). PhaseNet: a deep-neural-network-based seismic arrival-time picking method. Geophysical Journal International, 216(1), 261-273.

Zhu, W., McBrearty, I. W., Mousavi, S. M., Ellsworth, W. L., & Beroza, G. C. (2022). Earthquake phase association using a Bayesian Gaussian mixture model. Journal of Geophysical Research: Solid Earth, 127(5), e2021JB023249.